## 30 years of upper air soundings on board of R/V POLARSTERN

Amelie Driemel<sup>1</sup>, Bernd Loose<sup>1</sup>, Hannes Grobe<sup>1</sup>, Rainer Sieger<sup>1</sup>, and Gert König-Langlo<sup>1</sup> <sup>1</sup>Alfred-Wegener-Institut Helmholtz-Zentrum für Polar- und Meeresforschung, Bremerhaven, Germany *Correspondence to:* Gert König-Langlo (gert.koenig-langlo@awi.de)

**Abstract.** The research vessel and supply icebreaker POLARSTERN is the flagship of the Alfred-Wegener-Institut in Bremerhaven (Germany) and one of the infrastructural pillars of German Antarctic research. Since its commissioning in 1982, POLARSTERN has conducted 30 campaigns to Antarctica (157 legs, mostly austral summer), and 29 to the Arctic (94 legs, northern summer). Usually, POLARSTERN is more than 300 days per year in operation and crosses the Atlantic Ocean in

- a meridional section twice a year. The first radiosonde on POLARSTERN was released on the 29th of December 1982, two days after POLARSTERN started on its maiden voyage to the Antarctic. And these daily soundings have continued up to the present. Due to the fact that POLARSTERN has reliably and regularly been providing upper air observations from data sparse regions (oceans and polar regions), the radiosonde data are of special value for researchers and weather forecast services alike. In the course of 30 years (1982-12-29 to 2012-11-25) a total of 12378 radiosonde balloons were started on POLARSTERN.
- All radiosonde data can now be found at doi:10.1594/PANGAEA.810000. Each dataset contains the directly measured parameters air temperature, relative humidity and air pressure, and the derived altitude, wind direction and wind speed. 432 datasets additionally contain ozone measurements.

Although more sophisticated techniques (meteorological satellites, aircraft observation, remote sensing systems, etc.) have nowadays become increasingly important, the high vertical resolution and quality of radiosonde data remains paramount for 15 weather forecasts and modelling approaches.

## 1 Introduction

For more than 30 years now, the research vessel and supply icebreaker POLARSTERN is the flagship of the Alfred-Wegener-Institut in Bremerhaven (Germany). The commissioning of POLARSTERN on the 9th of December, 1982, was the result of a political decision to strengthen Germany's role in polar (at that time especially Antarctic) research: In 1978 the Federal Repub-

lic of Germany, represented by the Deutsche Forschungsgemeinschaft (DFG), became a member of the Scientific Committee on Antarctic Research (SCAR), and in 1980 the Alfred-Wegener-Institut for Polar Research was founded (Fütterer, 2007). R/V POLARSTERN was thus bound to become one of the infrastructural pillars of German Antarctic research. Home port of PO-LARSTERN was and still is Bremerhaven, Germany. Since its commissioning, POLARSTERN has conducted 30 campaigns to Antarctica (157 legs, mostly austral summer), and 29 to the Arctic (94 legs, northern summer)<sup>1</sup>. Usually, POLARSTERN is

<sup>&</sup>lt;sup>1</sup>The complete list of POLARSTERN campaigns can be found at http://www.pangaea.de/PHP/CruiseReports.php?b=Polarstern

# Science Science Science Science

more than 300 days per year in operation and crosses the Atlantic Ocean in a meridional section twice a year (Fütterer, 2007). It is therefore the perfect basis for upper air observations in data sparse regions (i.e. oceans and polar regions).

The first radiosonde on POLARSTERN was released on the 29th of December 1982, two days after POLARSTERN started on its maiden voyage to the Antarctic. Radiosondes are balloon-borne instruments which record atmospheric ("upper air")

- profile data, mostly temperature, humidity and pressure. The horizontal wind vector can be estimated when the displacement of the balloon is known. As DuBois (2002) wrote: "The contributions of this relatively simple device to the late twentieth-century way of life can hardly be exaggerated. No other factor contributed more to the systematization of weather observations, which is beneficial to all who depend upon meteorological prediction. [...]". Radiosondes are used extensively by weather forecast services, e.g. by the European Centre for Medium-Range Weather Forecasts (ECMWF). There are currently around 600-700
- land- and ship-based upper air observation stations routinely in operation worldwide which feed at least daily into the Global Telecommunication System (GTS) of the World Meteorological Organization. These data are used as input for routine weather forecast models such as the one run by the European Centre for Medium-Range Weather Forecast (ECMWF). However, Figure 1 shows, that ocean and polar regions are underrepresented (see also Dow (2004)). Regular radiosonde launches on ships therefore are of special value, and icebreakers like POLARSTERN can even provide data from the Antarctic coast and the
- North Pole.

In the following, we describe a data compilation of 30 years of upper air soundings on board of POLARSTERN using VAISALA radiosondes. We first provide details on the equipment used, the sampling procedure, and the parameters measured. Then, the data compilation is described in respect to extent, access and quality. A few examples of the data types are also given. In the last part we give a short overview of related radiosonde data in PANGAEA, and finish with a look into the future.

#### 20 2 Instrumentation, sampling, parameters

The upper air soundings on board of POLARSTERN were performed usually once a day at around 12 UTC. For this purpose the weather technician in charge released a helium filled balloon (TOTEX 600 g, 800 g, Japan) equipped with a radiosonde (RS-type, VAISALA, Finland) from the helideck of POLARSTERN at a height of 10 m above sea level. During strong wind conditions (>20 m/s), only 350 g balloons could be launched with a reasonable chance of success. All balloons were filled to reach an ascent velocity of 5 m/s; data recordings were made approximately every 25-50 m. The bursting point of the balloon, which set an end to the recordings, mostly was between 25 and 37 km height (König-Langlo, 2006). During the first years,

RS80 radiosondes were deployed. For a short time these were replaced by RS90 radiosondes. Since 2005-04-09 RS92-SGPW radiosondes have been in use, see Table 1 for details.

The radiosondes directly measured air pressure, air temperature and relative humidity. Until 1996, the wind vector was determined with the aid of the OMEGA navigation system, since then a GPS-based windfinding system has been used, leading to a remarkable improvement in the quality of the horizontal wind vector. Altitudinal information was calculated using the hydrostatic approximation.

The recorded data were transmitted from the sondes to POLARSTERN. Data reception and evaluation on POLARSTERN was carried out by a MicoCora (VAISALA, Finland) until 1996. After that the system was switched to a DigiCORA MW11, in 2003 to a DigiCORA MW21 and since 2012 a DigiCORA MW31 has been used (VAISALA, Finland).

During some cruises ozone profiles through the troposphere and the lower stratosphere were measured by connecting an ozonesonde (ECC-6A, ECC-6B, Science Pump Corporation, USA) to a normal radiosonde using an interface. Ozone concentration was measured by pumping air through a chemical solution and using the principal of iodide redox reaction to release electrons. 1500 g TOTEX balloons were used for these ascents. The DigiCORA was able to handle the data reception and evaluation of both, the normal radiosonde and the ozonesonde, simultaneously.

Together with hourly synoptical observations on board of POLARSTERN, radiosonde data were fed near real-time into the 10 GTS via the DWD (German weather service) and used routinely by various weather forecast services, such as the ECMWF. In the ECMWF "Monthly Monitoring Report" (available at http://www.ecmwf.int/en/forecasts/charts), e.g., you can find PO-LARSTERN meteorological data under the 'WMO Identifier' DBLK (which is the ship's call sign).

#### 3 Datasets, data access, examples, data quality, related data

#### 3.1 Datasets and data access

In total 12378 radiosonde balloons were started on POLARSTERN in the course of 30 years (from 1982-12-29 to 2012-11-25). All radiosonde data can now be found at doi:10.1594/PANGAEA.810000. The single radiosonde launches have been grouped into so-called "parents" according to the cruise leg they belong to (i.e. a parent contains all datasets of one cruise leg).

At doi:10.1594/PANGAEA.810000 you will find an overview table with information on the dates of the cruise leg, the number of single observations (i.e. launches), and links to the individual parents and respective cruise reports. Please note,

- that cruise labels for Antarctic cruises always begin with "ANT-", the ones for Arctic cruises with "ARK-". POLARSTERN radiosonde data for specific campaigns, areas or dates can also be searched by using the PANGAEA search engine (www. pangaea.de) and adding +PSradio<sup>2</sup> for a search in all parents, or by adding +PSradiosingle<sup>3</sup> for a search in all single radiosonde datasets ("childs"). Apart from the actual data (pressure, temperature, relative humidity, wind direction and wind speed according to altitude), each radiosonde dataset contains the information of when and where it was launched. Whenever
- possible, the height of the tropopause, the precipitable water content and the total ozone are also given. Furthermore, the central parent offers the link to the cruise report of the respective cruise leg (if available).

In Figure 2 all radiosonde launch sites of 30 years of POLARSTERN campaigns are mapped. The location of the homeport (Bremerhaven, North Sea) and the most frequent destinations of POLARSTERN (the Arctic and Neumayer Station, Weddell Sea) entail, that the radiosonde launch sites are mostly restricted to Atlantic Ocean regions. In terms of latitudinal coverage of the 12378 radiosonde launches Figure 3 confirms that high latitudes are indeed very well represented in our data.

<sup>&</sup>lt;sup>2</sup>Try for example *ozone* +*PSradio* to obtain all parents (=cruise legs) containing datasets with ozone data

<sup>&</sup>lt;sup>3</sup>Try for example *ozone* +*PSradiosingle* to obtain all radiosonde datasets containing ozone data

#### 3.2 Data examples for interested users

Of the 12378 single radiosonde launches, 432 contain ozone data. In Figure 5 the ozone data of POLARSTERN cruise ANT-XVII (launch sites see Figure 4) are plotted against altitude and latitude, irrespective of the longitude, date or time of measurement. Nevertheless, the distribution of stratospheric ozone, shaped by the Brewer-Dobson circulation, can be seen nicely.

Figure 6 shows an example of a meridional-height section of air temperature (again data from cruise ANT-XVII). Here, the low temperatures of the tropopause near the equator are apparent. With higher latitudes, the tropopause reaches lower altitudes and gets increasingly warmer (Lydolph, 1985).

Figure 7 shows an example of the wind velocity of the meridional-height section from cruise ANT-XVII. Pronounced maxima in the tropopause regions are clearly visible. They belong to the four westerly jet streams. In the northern hemisphere

the polar jet is clearly separated from the subtropical jet while in the southern hemisphere both jets coincide to be very close to each other.

Examples of practical applications of POLARSTERN radiosonde data can be found in various scientific articles, for example in Immler (2002), in John (2006) or in Yamazaki (2015) to mention just a few.

### 3.3 Data quality

The radiosonde data presented here have been measured by three different VAISALA sensors (RS80, RS90 and RS92) To optimize the data quality, the changes of the radiosonde types have to be taken into account. The data archived in PANGAEA represent the original values which were provided by the VAISALA reception and evaluation system. Except for the altitudinal information, which was calculated using the hydrostatic approximation, the data were not corrected for known systematic errors or biases. In the following, we provide an overview of the known biases and the respective correction proposals for

interested users.

RS80:

- The A-type HUMICAP humidity sensor is known to have a considerable dry bias mainly at low temperatures (Miloshevich, 2001). Correction methods have been proposed by Wang (2002).
- Also a time lag has been identified occurring due to the long response time of the humidity sensor at low temperatures.
  The proposed correction is to apply an algorithm published in Miloshevich (2004).
- In ice-supersaturated conditions ice-coating of the humidity sensor sometimes causes the measurements to stay near ice-saturation over large parts of the troposphere, not reflecting the conditions of the ambient air.

RS90:

The RS90 is equipped with two H-type HUMICAP humidity sensors that are alternately heated to remove condensed water 30 or ice, thus avoiding the effect of sensor icing.

A small time lag is still present, even though the H polymer has a faster sensor response time. Again, the according time constant in the correction algorithm of Miloshevich (2004) can be used to correct this.

Humidity data from all three VAISALA radiosondes suffer from the so-called daytime solar radiation dry bias (SRDB). Not being shielded against solar radiation, the sensors can be heated by incoming short-wave radiation during daytime launches (see e.g. Wang (2003) and Vömel (2007)). However, the RS92 operated together with the VAISALA DigiCORA Sounding System MW31 were ranked to be the world's top radiosonde at the 8th WMO Intercomparison of Radiosonde Systems in 2010 and GRUAN compatible (Nash, 2011).

5

#### 3.4 **Related upper air data in PANGAEA**

Radiosonde measurements on POLARSTERN are ongoing. Since 2012-11-25, which is the date of the last radiosonde launch discussed above, data of an additional 1160 POLARSTERN radiosonde launches have been archived in PANGAEA (status 2015-11-01). Apart from these, PANGAEA contains 11699 soundings performed at the Antarctic research station Neumayer

- and 9710 from the Arctic station Ny Ålesund (status 2015-11-01). Together with soundings from 28 other (non-AWI) stations 10 they are available in PANGAEA via the central archive of the Baseline Surface Radiation Network (BSRN). For an overview of available BSRN radiosonde data (monthly files) see http://www.pangaea.de/PHP/BSRN\_Status.php?q=LR1100. Historical data on radiosonde launches from the Georg Forster Station were described in König-Langlo (2009) and can be accessed in PANGAEA via doi:10.1594/PANGAEA.547983. Furthermore, two large re-analysis datasets were archived in PANGAEA
- for publications by Stickler (2014) and Ramella-Pralungo (2014), respectively. From the german research vessels SONNE, 15 METEOR (1964) and METEOR (1986) PANGAEA offers 61, 1510 and 88 datasets from radiosonde launches, respectively.

#### 4 Conclusions

Even though more sophisticated techniques (meteorological satellites, aircraft observation, remote sensing systems, etc.) have become increasingly important in weather forecasts, the high vertical resolution and quality of radiosonde data remains crucial for weather forecast quality, for detecting changes in climate, and for the validation of modelling and satellite products (Dow, 20 2004). The Alfred Wegener Institute is committed to provide high-quality and open access research data to the scientific community and the interested public, and will thus continue with upper air soundings (and other under-way measurements) on board of POLARSTERN until its decommissioning. The new research vessel POLARSTERN II, which according to the current schedule will be handed over to the scientific community in 2019, will most likely continue the work of over 30 years 25 of dedicated POLARSTERN meteorology.

Acknowledgements. We would like to thank Wolfgang Cohrs for the technical assistance and the creation of Figure 2. Special thanks to all DWD weather technicians on board of POLARSTERN responsible for launching the sondes: Hans Ohlendorf, Dieter Bassek, Herbert Köhler, Wolf-Thilo Ochsenhirt, Juliane Schostak, Hartmut Sonnabend, Roland Menzel, Hans-Peter Lambert, Klaus Buldt, Edmund Knuth, Thorsten Truscheit and Juliane Hempelt.

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
