# Peer review of "years of upper air soundings on board of R/V POLARSTERN"

_Earth System Science Data, 2015_

## Referee Comment (RC1) · Anonymous Referee #1 · 13 Apr 2016

General Comments: This manuscript does a great job detailing the 30 years dataset of upper air soundings on board of R/V POLARSTERN. The short article is an excellent source of metadata for those looking to use this dataset in the future. Section 3 was particularly useful due to the scientific examples, and the details about the data quality of the entirety of the dataset. The dataset will be particularly useful for Arctic and Antarctic researcher looking to study the upper atmosphere. It was great to be able to easily understand and find the dataset in PANGAEA. The data quality and the metadata in PANGAEA makes downloading pieces of the dataset effortless.

Specific Comments: 1. Section 3.4 doesn't seem unique and necessary in this kind of article. This might because I'm not too familiar with science data papers. Thus, I'm not certain that section is needed to support the thesis of this article.

2. On page 3 line 11 there is a reference to the ECMWF links. I wasn't able to navigate that page to try and see what was being reference. Clarity with this link and explain would make the connection to forecast models and soundings stronger.

3. On page 3 line 17 there is mention about the how the parents contain all datasets of one cruise leg. Here might be a good location for add the number of cruises/parents in this dataset.

Technical Comments: 1. Table 1 caption change to "Overview of the radiosonde types on board POLARSTERN"

---

## Referee Comment (RC2) · Anonymous Referee #2 · 17 May 2016

Major comments: The manuscript describes a 30-year radiosonde dataset collected from R/V POLARSTERN. The dataset would be very useful for various scientific applications. My main concerns are lack of more detailed information on the instruments and some statistics on the characteristics of the dataset, and overall data quality. The users very often trust the data creators and use the data as the "truth". It is the responsibility of the data creators to conduct some rudimentary quality control to the data and make the users aware of potential issues. I provide some details in "Specific comments". I think that the authors should put themselves in the users' shoes to think about what additional information should be provided. Based on my evaluation, I think that the manuscript is appropriate for ESSD, but needs some revisions before it is ready for publications.

Specific comments: 1. P2, L12: ECMWF was defined before.

2. P2, L25, "every 25-50m", Why is the vertical resolution only 25-50m? Does it contain high resolution (1 or 2 sec) or just GTS data? The latter only has data available at standard and significant pressure levels. Please clarify this. For research purpose, the high resolution data would be very useful.

3. P2, L28, Table 1: Please clarify whether it is RS80A or RS80H. It would be useful to list the sensor types in the table and their accuracy from the manufacture.

4. P3, L24: Besides "when" "where", does it contain the radiosonde type used? This would be very useful metadata information.

5. P3, Fig. 2: I would recommend that you use different colors representing the years when the data were collected if it is not too messy.

6. P3, Fig. 3: It might be useful to make similar statistics as a function of years.

7. P4, Section 3.3: Did you apply any basic data quality control procedures to all soundings, such as the limit, outlier, and monotonic pressure tests, to remove or flag any gross errors in the data?

8. P4, references: Wang (2002) should be Wang et al. (2002), and Miloshevich (2004) should be Miloshevich et al. (2004). This applies to all citations.

9. P5, L3: Add the reference Wang et al. (2013). Wang, J., L. Zhang, A. Dai, F. Immler, M. Sommer and H. Voemel, 2013: Radiation dry bias correction of Vaisala RS92 humidity data and its impacts on historical radiosonde data. J. Atmos. Oceanic Technol., 30, 197-214.

10. P4, Section 3.3: In addition to the biases listed here, the ship soundings might contain ship-specific biases, such as ship deck heating/cooling biases discussed in Ciesielski et al. (2014). They should be mentioned. Ciesielski, P. E., H. Yu, R. H. Johnson, K. Yoneyama, M. Katsumata, C. N. Long, J. Wang and others, 2014: Quality-controlled upper-air sounding dataset for DYNAMO/CINDY/AMIE: Development and corrections. J. Atmos. Oceanic Technol., 31, 741-764.

---

## Author Comment (AC1) · 20 May 2016

First of all, we are very grateful for the constructive comments of both reviewers. We are convinced that by heeding the reviewers' recommendations we could greatly improve the manuscript. Below, we have addressed all issues indicated in the reviewers' reports, and we believe that the revised version can meet the publication requirements of ESSD.

General comments:

**COMMENT 1: Section 3.4 doesn't seem unique and necessary in this kind of article. This might because I'm not too familiar with science data papers. Thus, I'm not certain that section is needed to support the thesis of this article. #RESPONSE: The reviewer is right in the respect that this section is not needed to support the main focus of**

the article. Nevertheless, we would like to keep it, although we have shortened it considerably. In our opinion, this section is a good and important means to shortly address similar or complementing datasets (similar to a literature review of a scientific article).

**COMMENT 2: On page 3 line 11 there is a reference to the ECMWF links. I wasn't able to navigate that page to try and see what was being reference. Clarity with this link and explain would make the connection to forecast models and soundings stronger. #RESPONSE: We are sorry for that, the idea was to keep the link short. But the page really is confusing. We therefore changed the link to the one directly leading to the reports: http://www.ecmwf.int/en/forecasts/quality-our-forecasts/monitoring-observing-system/ecmwf-global-data-monitoring-report-archive**

**COMMENT 3: On page 3 line 17 there is mention about the how the parents contain all datasets of one cruise leg. Here might be a good location for add the number of cruises/parents in this dataset. #RESPONSE: This is a very good idea, number of cruises (legs) have been added, the sentence now reads: "The single radiosonde launches have been grouped into so-called "parents" according to the cruise leg they belong to (i.e. a parent contains all datasets of one cruise leg, all in all 210 legs = parents)."**

Technical Comments:

**COMMENT 1: Table 1 caption change to "Overview of the radiosonde types on board POLARSTERN" #RESPONSE: Caption was changed accordingly.**

---

## Author Comment (AC2) · 20 May 2016

First of all, we are very grateful for the constructive comments of both reviewers. We are convinced that by heeding the reviewers' recommendations we could greatly improve the manuscript. Below, we have addressed all issues indicated in the reviewers' reports, and we believe that the revised version can meet the publication requirements of ESSD.

General:

Major comments: [..]. My main concerns are lack of more detailed information on the instruments and some statistics on the characteristics of the dataset, and overall data quality. The users very often trust the data creators and use the data as the "truth". It is the responsibility of the data creators to conduct some rudimentary quality control

to the data and make the users aware of potential issues. I provide some details in "Specific comments". I think that the authors should put themselves in the users' shoes to think about what additional information should be provided. Based on my evaluation, I think that the manuscript is appropriate for ESSD, but needs some revisions before it is ready for publications.

Specific comments:

**COMMENT 1. P2, L12: ECMWF was defined before. #RESPONSE: We deleted the repeated definition**

**COMMENT 2. P2, L25, "every 25-50m", Why is the vertical resolution only 25-50m? Does it contain high resolution (1 or 2 sec) or just GTS data? The latter only has data available at standard and significant pressure levels. Please clarify this. For research purpose, the high resolution data would be very useful. #RESPONSE: Until 1998-06-06 the profile data where taken every 10 seconds which results in a vertical resolution of about 50 m. Later, the sampling rate was increased to 5 seconds. A subset of these data - containing only the standard and significant pressure levels according to the TEMP definition of the WMO (Manual on Codes International Codes, VOLUME I.1, Part A - Alphanumeric Codes, WMO-No. 306(1995 edition)) was transferred into the Global Telecommunication System (GTS) without delay. We added this information (first two sentences of this response) to the respective paragraph.**

**COMMENT 3. P2, L28, Table 1: Please clarify whether it is RS80A or RS80H. It would be useful to list the sensor types in the table and their accuracy from the manufacture. #RESPONSE: We first used RS80-15N with the Omega wind finding technique. From 1996 on, when GPS came into use, we used RS80-15G. Both sondes were equipped with RS80A-Humicap radiosonde humidity sensor as described in Chapter 3.3. We have added this information also in Chapter 2. Since we know about the considerable dry bias at low temperatures and the time lag of this sensors we added Chapter 3.3 including links to the corresponding literature. In our opinion Chaper 3.3 is more informative than adding the accuracy of the sensors given from the manufacturer to Table 1.**

**COMMENT 4. P3, L24: Besides "when" "where", does it contain the radiosonde type used? This would be very useful metadata information. #RESPONSE: Yes you are right, we added a column to https://doi.pangaea.de/10.1594/PANGAEA.810000 with information on which sonde was used during which cruise leg.**

**COMMENT 5: P3, Fig. 2: I would recommend that you use different colors representing the years when the data were collected if it is not too messy. #RESPONSE: In PANGAEA each Campaign comes with its own map. Thus, the customer of the dataset has a perfect option to see when and where measurements have been taken. The idea of Fig. 2 is just to show where the measurements took place typically. Coloring 30 years with different colors would definitely overload Fig. 2.**

**COMMENT 6: P3, Fig. 3: It might be useful to make similar statistics as a function of years. #RESPONSE: We added a second part to the figure with the statistics according to years.**

**COMMENT 7: P4, Section 3.3: Did you apply any basic data quality control procedures to all soundings, such as the limit, outlier, and monotonic pressure tests, to remove or flag any gross errors in the data? #RESPONSE: Yes. We have validation routines which have been applied to all soundings. Of course only internal inconsistencies and physical impossible data can be checked. The data reception systems described in Chapter 2 follow about the same ideas. Thus, we had to correct (exclude) only very few data. We added this information including the information on the general checks applied at the beginning of Section 3.3.**

**COMMENT 8: P4, references: Wang (2002) should be Wang et al. (2002), and Miloshevich (2004) should be Miloshevich et al. (2004). This applies to all citations. #RESPONSE: Yes, sorry for that. We corrected it in all citations.**
**COMMENT 9: P5, L3: Add the reference Wang et al. (2013). Wang, J., L. Zhang, A. Dai, F. Immler, M. Sommer and H. Voemel, 2013: Radiation dry bias correction of Vaisala RS92 humidity data and its impacts on historical radiosonde data. J. Atmos. Oceanic Technol., 30, 197-214. #RESPONSE: Citation has been added, thanks for the hint.**

**COMMENT 10: P4, Section 3.3: In addition to the biases listed here, the ship soundings might contain ship-specific biases, such as ship deck heating/cooling biases discussed in Ciesielski et al. (2014). They should be mentioned. Ciesielski, P. E., H. Yu, R. H. Johnson, K. Yoneyama, M. Katsumata, C. N. Long, J. Wang and others, 2014: Quality-controlled upper-air sounding dataset for DYNAMO/CINDY/AMIE: Development and corrections. J. Atmos. Oceanic Technol., 31, 741-764. #RESPONSE: Again, many thanks for this hint. We cited this paper accordingly and added a paragraph at the end of section 3.3 elaborating on the ship-specific biases we know from POLARSTERN and our means to avoid them. The paragraph reads: "Another source of biases can be ship-specific influences such as heating/cooling biases discussed in Ciesielski et al. (2014). As already stated in section 2, all soundings on board of POLARSTERN were started from the helideck at a height of 10 m above sea level, resulting in datasets beginning at an altitude of 10 m a.s.l.. The values for this first data point of radiosonde launches normally are taken from ground measurements. Due to the fact, that no unbiased ground measurements can be taken on a ship, the (largely unbiased) values of the ship's Luv-sensors (29 m a.s.l. for temperature and humidity, 39 m a.s.l. for wind) were used. The air pressure value for this first data point was calculated from the ship's meteorological observatory (16 m a.s.l.). The pressure sensor has an external inlet which ends in a labyrinth in the ship's mast to reduce ship-specific influences. Beginning with the second data point (altitude level), the data originate from the radiosonde itself. In the rare case that the quality check revealed ongoing disturbances for the second data point (e.g. by the ship's plume), the respective data was deleted from the records. For the third altitude level onwards, no ship-based biases were expected anymore."**

[Figure]

[Figure]

Fig. 1. Figure number 3 with added information on launches vs. year